# Structural insights into trans-histone regulation of H3K4 methylation by unique histone H4 binding of MLL3/4

Yanli Liu[1], Su Qin[1], Tsai-Yu Chen[2], Ming Lei[1], Shilpa S. Dhar[2], Jolene Caifeng Ho[1], Aiping Dong[1], Peter Loppnau[1], Yanjun Li [1], Min Gyu Lee[2] & Jinrong Min[1,3]

MLL3 and MLL4 are two closely related members of the SET1/MLL family of histone H3K4 methyltransferases and are responsible for monomethylating histone H3K4 on enhancers, which are essential in regulating cell-type-specific gene expression. Mutations of MLL3 or MLL4 have been reported in different types of cancer. Recently, the PHD domains of MLL3/4 have been reported to recruit the MLL3/4 complexes to their target genes by binding to histone H4 during the NT2/D1 stem cell differentiation. Here we show that an extended PHD domain (ePHD$_6$) involving the sixth PHD domain and its preceding zinc finger in MLL3 and MLL4 specifically recognizes an H4H18-containing histone H4 fragment and that modifications of residues surrounding H4H18 modulate H4 binding to MLL3/4. Our in vitro methyltransferase assays and cellular experiments further reveal that the interaction between ePHD$_6$ of MLL3/4 and histone H4 is required for their nucleosomal methylation activity and MLL4-mediated neuronal differentiation of NT2/D1 cells.

[1] Structural Genomics Consortium, University of Toronto, 101 College Street, Toronto, Ontario M5G 1L7, Canada. [2] Department of Molecular and Cellular Oncology, The University of Texas MD Anderson Cancer Center, 1515 Holcombe Boulevard, Houston, Texas 77030, USA. [3] Department of Physiology, University of Toronto, Toronto, Ontario M5S 1A8, Canada. These authors contributed equally: Yanli Liu, Su Qin, Tsai-Yu Chen, Ming Lei. Correspondence and requests for materials should be addressed to M.G.L. (email: mglee@mdanderson.org) or to J.M. (email: jr.min@utoronto.ca)

During early embryonic development, development-regulatory genes need to be precisely turned on or off in order to form a complex, multi-tissue, and multi-organ organism. Trithorax group (trxG) and Polycomb group (PcG) proteins were initially discovered in *Drosophila* to control body-plan formation, and were later found to be conserved in mammals and implicated in development, stem cell biology, and cancers[1,2]. The trithorax and PcG of proteins function as positive and negative epigenetic regulators, respectively, and maintain the transcriptional states of development-regulatory genes, such as the *Hox* genes, which are established in early embryonic development.

The trithorax family of histone H3 lysine 4 (H3K4) methyltransferases is conserved from yeast to human. This family of H3K4 methyltransferases is also named as the SET1/MLL (also called COMPASS) family after its founding member SET1, which is first identified in yeast[3]. The SET1/MLL family of methyltransferases function as multi-component complexes, and there exist at least six SET1/MLL complexes (MLL1–4, SET1A, and SET1B) in mammals. In recent years, tremendous progress has been made in understanding the functional diversity of these different SET1/MLL complexes in vitro and in vivo. For instance, *Drosophila* Set1 and its mammalian homologs SET1A/B have been shown to be responsible for the bulk levels of H3K4me2 and H3K4me3 in cells[4]. MLL1 is responsible for H3K4 trimethylation at less than 5% of gene promoters, including *Hox* genes, in mouse embryonic fibroblasts[5], whereas its close homolog MLL2 lays down H3K4 trimethylation mark at bivalently marked gene promoters in mouse embryonic stem (ES) cells[6]. MLL3/4 and their *Drosophila* homolog Trr are the major H3K4me1 methyltransferases responsible for H3K4me1 modifications on enhancers, which are essential in regulating cell-type-specific gene expression[4,7–12]. Although these different SET1/MLL complexes share several core subunits (e.g., WDR5, RBBP5, ASH2L, and DPY-30), other unique components/domains in these complexes would play critical roles in coordinating division of labor among these SET1/MLL family members in regulation and recruitment of the SET1/MLL family of methyltransferase complexes[3,13]. For instance, a unique PHD domain in MLL1 was revealed to recognize the histone H3K4me3 mark, and this binding is critical for MLL1-dependent target gene expression[14,15]. PSIP1, which is a histone H3K36me3 binder, preferentially associates with the MLL2 complex[13]. CFP1, a unique component in the SET1A/B complexes[13,16], has been known to selectively bind to non-methylated CpGs in vitro and in vivo[17–21], and target the SET1A/B complexes to its target chromatin regions[17]. Therefore, the different SET1/MLL complexes are recruited to distinct genomic loci through specific recruiting mechanisms involving their unique components.

MLL3 (also called KMT2C) and MLL4 (alias KMT2D and ALR) are two closely related members of the SET1/MLL family of histone H3K4 methyltransferases, and often act as tumor suppressors[22,23]. Mutations of MLL3 or MLL4 have been frequently found in patients of Kabuki syndrome, childhood medulloblastoma, acute myeloid leukemia, and lymphomas[22,24–27]. Both MLL3 and MLL4 methyltransferase complexes regulate diverse metabolic processes including circadian control of bile acid homeostasis[28]. Both MLL4 and MLL3 (to a lesser extent) also play a critical role in differentiating NT2/D1 stem cells by activating differentiation-specific genes in a histone H3K4me3-dependent manner[29]. Intriguingly, the sixth PHD domain of MLL4 (PHD$_6$) was found to be required to recognize unmethylated histone H4 N-terminal tail and this binding ability is essential for MLL4's histone H3K4 methylation activity and MLL4-mediated cellular differentiation[29]. In contrast, Chauhan et al. reported that the PHD$_6$ domain in MLL4 or the corresponding domain in the

*Drosophila* LPT (Lost PHD domains of Trr) bound to both unmodified histone H3K4 and H3K4me1/2[30]. Considering the importance of MLL3/4 in regulating enhancer transcription and gene expression, and their implication in various cancers and other diseases, it is of great interest to study how the PHD domains of MLL3/4 recognize histones and recruit the MLL3/4 methylation activity to the target chromatin regions.

In this study, we set out to clarify the histone binding nature of the PHD$_6$ domain in MLL3/4, and find that an extended PHD domain including the PHD$_6$ domain and the preceding zinc finger (ePHD$_6$) in MLL3/4 specifically recognizes an H4H18-containing fragment of histone H4. Our complex structure provides structural insights into how this extended PHD domain specifically recognizes the H4H18-containing fragment of histone H4. Furthermore, our in vitro methyltransferase assays using recombinant nucleosomes as substrate reveal that the interaction between ePHD$_6$ of MLL3/4 and histone H4 is required for their nucleosomal methylation activity. Our cellular experiments also indicate that the binding activity of ePHD$_6$ is required for MLL4-mediated neuronal differentiation of NT2/D1 cells. Thus, our study identifies a binding mode of the extended PHD domain of MLL3/4 for the histone H4 N-terminal fragment and provides insights into a trans-histone regulatory mechanism of MLL3/4-mediated H3K4 methylation.

## Results

**MLL3/4 ePHD$_6$ recognizes histone H4H18-containing fragment.** Both MLL3 and MLL4 are multi-domain proteins, containing not only the catalytic SET domain but also a few other domains, including two clusters of PHD domains (Fig. 1a). By peptide pull-down assays, it has been shown that the fourth (PHD$_4$), fifth (PHD$_5$), and sixth (PHD$_6$) PHD domains of MLL4 are able to recognize a histone H4 fragment containing the first 23 amino acids, either unmethylated (H4R3me0) or asymmetrically dimethylated on histone H4R3 (H4R3me2a), whereas symmetrical dimethylation of histone H4R3 (H4R3me2s) disrupts the binding[29]. The interaction between histone H4$_{1-23}$ and PHD$_6$ domain has also been confirmed by Isothermal titration calorimetry (ITC)[29]. To better understand the binding specificity of these PHD domains in MLL3/4, we cloned the three PHD domains (PHD$_{4-6}$) either individually or in combinations for both MLL3 and MLL4, but we were only able to obtain soluble and stable proteins for PHD$_5$, PHD$_6$, and PHD$_{5-6}$ of MLL3 and PHD$_5$ and PHD$_6$ of MLL4. Of note, a zinc finger precedes the PHD$_6$ domain in both MLL3 and MLL4 and the PHD$_3$ domain in MLL3 (Fig. 1a). So we also cloned the fragments of MLL3 and MLL4 covering both the zinc finger and the PHD domain (hereinafter referred to as extended PHD (ePHD) domain), which were also soluble and stable.

Our ITC studies revealed that the ePHD$_6$ domain of neither MLL3 nor MLL4 displayed detectable binding to the histone H4 peptide covering the first 12 residues (H4$_{1-12}$) no matter whether it is unmethylated (H4R3me0) or asymmetrically methylated (H4R3me2a) (Table 1, Supplementary Fig. 1). Instead, the ePHD$_6$ domains of MLL3 and MLL4 bound to a histone H4 fragment covering the residues 11–21 (H4$_{11-21}$) (Table 1). However, the ePHD$_6$ domains of MLL3 and MLL4 exhibited slightly higher binding affinity to the H4$_{1-24}$ peptide compared to the H4$_{11-21}$ peptide, and the R3A mutant of the histone H4$_{1-24}$ peptide also displayed reduced binding affinity to ePHD$_6$ of MLL3 and MLL4 by 2 or 3 folds, respectively, indicating that the H4R3-containing fragment somewhat may contribute to the ePHD$_6$ binding, which will be discussed further later on (Table 1, Supplementary Fig. 1). Both MLL3 and MLL4 showed no detectable binding to the H3K4 peptides regardless of its methylation status (Table 1,

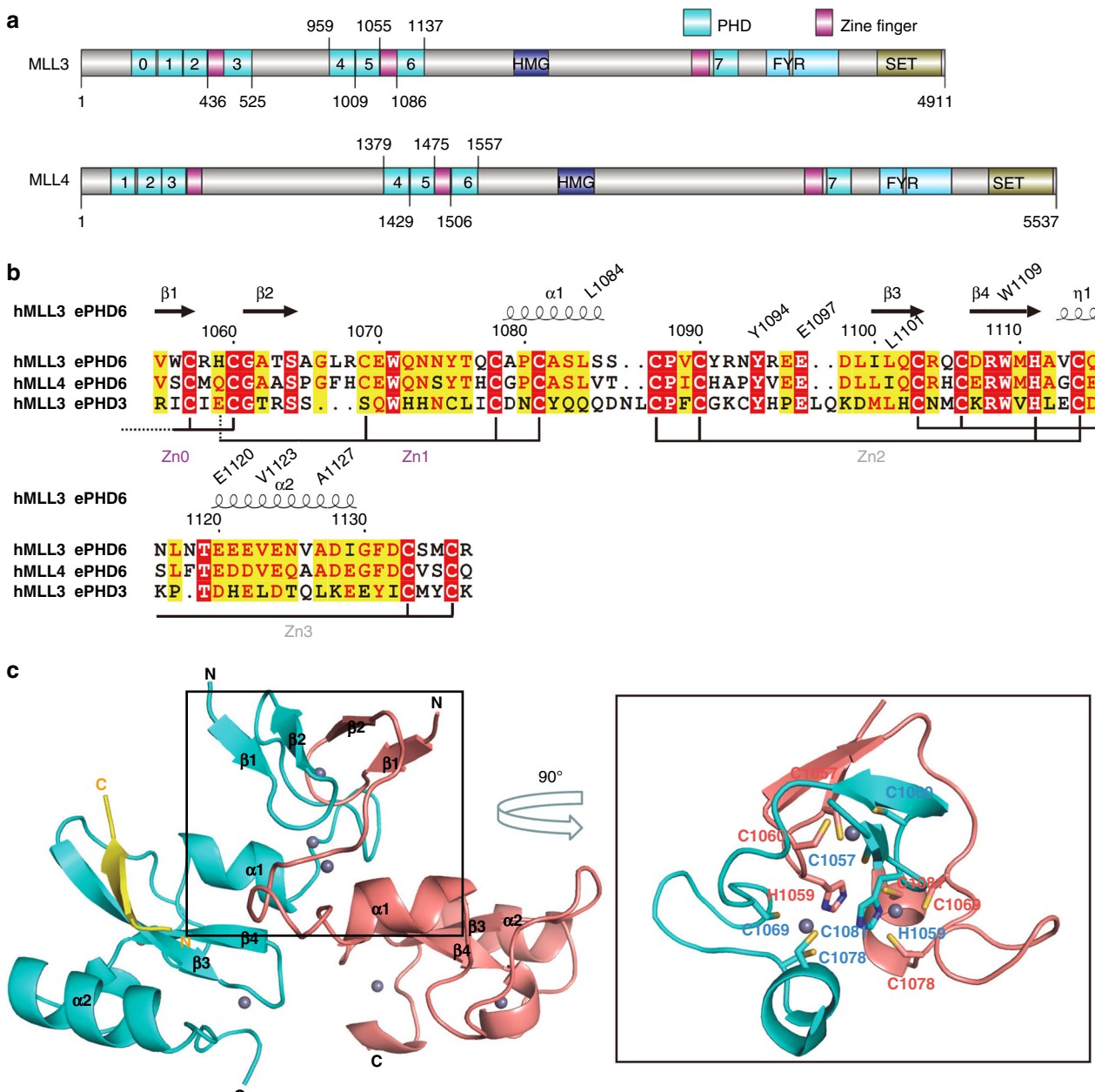

**Fig. 1** Overall structure of the extended PHD6 domain of MLL3 in complex with the H4H18 peptide. **a** Domain structure of human MLL3 and MLL4. PHD: plant homeodomain-linked zinc finger; SET: the histone methylation catalytic domain; HMG: high-mobility group; FYR: phenylalanine and tyrosine rich domain. **b** Structure-based sequence alignment of the extended PHD domains 3 and 6 of MLL3 and the extended PHD domain 6 of MLL4. Secondary structure elements and residue numbers of the ePHD6 domain of MLL3 are indicated above the sequence alignment. The alignments were constructed with ClustalW[56] and refined with ESPript[57]. **c** Overall structure of the ePHD6 domain of MLL3 in complex with H4$_{1-20}$ peptide. The two molecules of ePHD6 domains are colored in cyan and salmon, respectively. Structure figures were generated by using PyMOL

Supplementary Fig. 1). Furthermore, our binding results also showed that only the extended PHD6 domain (ePHD6) displayed binding to the histone H4 peptide for both MLL3 and MLL4 (Table 2, Supplementary Fig. 2). The PHD6 domain itself of MLL3/4 was not sufficient for binding to H4, reminiscent of the binding of arginine methylated PIWI proteins by the extended Tudor domain of SND1, where the Tudor domain itself is also not sufficient for binding[31].

**Complex structure of MLL3 ePHD6 and a histone H4H18 peptide.** In order to elucidate the molecular mechanism of the specific recognition of the histone H4 peptide by the extended

PHD domains of MLL3/4, we crystallized the MLL3 ePHD6 domain in complex with the histone H4$_{1-20}$ peptide after trying different MLL3/4 constructs and histone H4 peptides of various lengths and modifications, and determined its crystal structure (Table 3). The crystal structure of the MLL3 ePHD6 domain revealed that it contains a canonical PHD domain preceded by a zinc knuckle composed of two antiparallel β strands and an α helix (Fig. 1b). The MLL3 ePHD6 domain formed a dimer through two N-terminal cysteine residues, which coordinated a zinc ion (Zn0) with two cysteine residues from each protomer of the dimer (Fig. 1c). In addition, a domain swapping between these two ePHD6 molecules also contributed to the dimer formation, in which a zinc ion (Zn1) was chelated by three cysteine

**Table 1 Binding affinities of different histone peptides to the ePHD$_6$ of MLL3 and MLL4 measured by ITC**

| Histone peptides | $K_d$ (μM) | |
|---|---|---|
| | MLL3_ (ePHD$_6$)$_{1055-1144}$ | MLL4_ (ePHD$_6$)$_{1475-1564}$ |
| H4$_{1-24}$ | 23 ± 3 | 13 ± 1 |
| H4$_{1-12}$ | NB | NB |
| H4$_{1-12}$R3me2a | NB | NB |
| H4$_{11-21}$ | 35 ± 2 | 52 ± 2 |
| H4$_{1-24}$R3A | 66 ± 3 | 28 ± 1 |
| H3$_{1-20}$ | NB | NB |
| H3$_{1-15}$K4me2 | NB | NB |
| H3$_{1-15}$K4me3 | NB | NB |

$K_d$ values were calculated from single measurement and errors were estimated from fitting curve by Origin software package
NB, no detectable binding

residues from one molecule and one histidine residue from the other molecule (Fig. 1c). The dimer involved only the preceding zinc fingers, but not the PHD domain itself. However, our gel-filtration profiles showed that the extended PHD$_6$ domain of MLL3 behaved as a monomer in solution, like the PHD$_6$ domain itself (Supplementary Fig. 3). Therefore, the dimer was formed probably due to artificial crystal packing, which has been observed in other PHD proteins[32,33].

In the complex structure, the histone H4 peptide bound to one molecule of the two ePHD$_6$ molecules and was induced to form a β strand antiparalleling with β3 of ePHD$_6$. Of note, the β3 of ePHD$_6$ was also induced to become longer than that in the apo molecule (Fig. 2a). The residues $_{14}$GAKRHR$_{19}$ of the histone H4 peptide could be traced reliably (Fig. 2b). The H4G14, H4K16, and H4H18 residues formed main chain hydrogen bonds with the main chains of Q1102, I1100, E1097, and D1098 of MLL3, respectively, allowing the β-sheet formation between the H4 peptide and ePHD$_6$ of MLL3 (Fig. 2b). The side chain of H4A15 was accommodated in a shallow hydrophobic pocket, formed by L1101, V1123, and A1127 of MLL3, while its main chain was restricted by W1109 of MLL3 (Fig. 2c, d). The aliphatic chain of H4K16 stacked with the aromatic ring structure of W1109 and interacted with the hydrophobic side chain of I1100. H4R17 formed a salt bridge with E1120 and H4R19 formed a salt bridge with E1097 (Fig. 2d, e). In addition to the salt bridge with E1120, H4R17 also formed a hydrogen bond with the main chain of E1097. Thus the positive side chain of H4R17 was clasped in a negative channel formed by E1120 and E1097. The imidazole ring of H4H18 lied against the side chain of Y1094 and was further restricted by the main chain carbonyl groups of L1084 from the preceding zinc finger (Zn1, aa 1055 to 1085) and R1095 of PHD$_6$ (Fig. 2d, f). The α-helix α1 from the zinc finger Zn1, which packed against the β4 of ePHD$_6$, contributed to the formation of the peptide binding groove (Fig. 2b). This also explains why the PHD$_6$ domain alone lacked the binding ability to histone H4, because it required at least the α-helix α1 of the zinc finger Zn1 to retain the histone H4-binding ability (Table 2).

To validate the importance of the interacting residues observed from the complex structure, we introduced point mutations into MLL3 for binding studies. Mutating W1109 of MLL3 to alanine disrupted its binding to histone H4, underscoring its importance in recognizing H4K16 and H4A15 (Fig. 2g). Mutating E1120 reduced its binding to histone H4 significantly and E1097A mutation reduced its binding to histone H4 about 2 folds, respectively, presumably due to the loss of the salt bridge interactions with the H4R17 and H4R19 residue, respectively (Fig. 2g). Mutating Y1094 led the protein to become insoluble,

implying that Y1094 also played a structural role in addition to interacting with H4H18.

To further explore the importance of the interacting residues within the histone peptide, we synthesized a series of histone H4-derived mutant peptides and measured their binding affinities to ePHD$_6$ of MLL3 and MLL4 by ITC (Fig. 2h, Supplementary Fig. 4). Consistent with our structural studies, mutation of R17 or H18 to alanine disrupted the binding totally, and the K16A and R19A mutations weakened the binding significantly. The H4R17 residue could be methylated by PRMT7[34,35], and the PRMT7-mediated H4 methylation has been shown to hamper its binding to the PHD domains of MLL4, and repress MLL4 target genes[29]. Our ITC binding results showed that methylation of H4R17 indeed reduced its binding to the ePHD$_6$ domains of MLL3/4 to different extents depending on the methylation status, i.e., monomethylation slightly reduced the binding while asymmetrical and symmetrical dimethylation almost abolished the binding (Fig. 2h, Supplementary Fig. 4). This may be because the asymmetrical or symmetrical dimethylation of H4R17 might disrupt its hydrogen bonding or salt bridge interactions with E1097 and E1120. Trimethylation of H4K16 showed different effects on binding to MLL3 and MLL4, which will be discussed in the following section. Taken together, our structural and binding studies suggest that the extended PHD domain specifically recognized the H4H18-containing sequence of histone H4.

**Structural comparison to other PHD domains.** Structural comparison to the histone H3K4 binding PHD domains of BPTF[36], BHC80[37], MLL5[38], and UHRF1[33] reveals that all of these published PHD domains utilize an enclosed binding pocket to recognize the free N-terminal amine group of residue A1 of histone H3, whereas the MLL3 PHD domain has an open binding groove, which explains why the MLL3 PHD domain is able to bind to a sequence motif in the middle of the histone H4 N-terminal tail (Supplementary Figs. 5 and 6). When we modeled the histone H3K4 peptide into the MLL3 structure, we found that H3A1 and H3R2 would clash with α2 and Q1102 of MLL3 (Supplementary Figs. 5 and 6), respectively, explaining why the extended PHD domain of MLL3/4 could not bind to the H3K4 peptides, consistent with our binding data (Table 1).

As far as we know, all the H3K4me3-binding PHD domains harbor an invariant W residue to recognize the tri-methylated K4. Interestingly, in our structure, a W (W1109) residue is also present in the same position as those PHD domains, which might form a tri-methyl lysine-binding pocket of just one aromatic residue W similar to that of the MLL5 PHD domain[38]. In MLL5, its PHD domain recognizes the H3K4me3 residue by a pocket formed by a tryptophan and a methionine. In our structure, H4K16 was surrounded by W1109 and I1100. H4K16 methylation was recently identified in mouse brain[34]. We were curious if the H4K16 methylation would enhance its binding to this single W pocket. Our binding results showed that the H4K16 methylation did enhance binding to MLL4, but not to MLL3 (Fig. 2h, Supplementary Fig. 7). Sequence and structural analysis of MLL3 and MLL4 revealed that the isoleucine residue I1100, which was located in the potential methyl lysine-binding pocket in MLL3, is a leucine residue in the corresponding position (L1520) in MLL4 (Fig. 1b). Leucine has been reported to form part of an aromatic cage for methylation recognition, such as in mutated ZCWPW2 and L3MBTL1[39,40], whereas to our best knowledge isoleucine has not been reported as an aromatic cage forming residue. Consistently, when we mutated I1100 of MLL3 to leucine, the mutant exhibited enhanced binding affinity to the H4K16me3 peptide, while the L1520I mutant of MLL4 showed reduced binding affinity (Supplementary Fig. 7). Therefore,

although the ePHD$_6$ domains of both MLL3 and MLL4 bound to the H4H18-containing fragment of H4, the modification of the surrounding residues might distinguish their binding ability, specifically, H4K16 methylation increased its binding to MLL4, whereas weakened its binding to MLL3.

**H3K4 methylation activity of MLL3/4 requires binding to H4**. To address the trans-histone regulation of H3K4 methylation by histone H4, we next examined how the binding of ePHD$_6$ of MLL3/4 to the histone H4H18 fragment affects the MLL3/4-catalyzed H3K4 methylation in vitro using recombinant nucleosomes as substrates. We made several truncation fusion proteins of both MLL3 and MLL4, and also made mutated ePHD$_6$ domains of MLL3/4 containing mutations in some key residues

**Table 2 Binding affinities of different MLL3 or MLL4 constructs to the histone H4 peptide H4$_{11-21}$**

| MLL3/4 constructs | H4$_{11-21}$ peptide $K_d$ (μM) |
|---|---|
| MLL3_(ePHD$_6$)$_{1055-1144}$ | 35 ± 2 |
| MLL3_(ePHD$_6$)$_{1075-1144}$ | 31 ± 4 |
| MLL3_(PHD$_6$)$_{1085-1144}$ | Weak binding[a] |
| MLL3_(PHD$_{5-6}$)$_{1008-1144}$ | 83 ± 14 |
| MLL3_(PHD$_5$)$_{1009-1055}$ | NB |
| MLL3_(ePHD$_3$)$_{436-525}$ | NB |
| MLL4_(ePHD$_6$)$_{1475-1564}$ | 52 ± 2 |
| MLL4_(ePHD$_6$)$_{1495-1564}$ | 48 ± 5 |
| MLL4_(PHD$_6$)$_{1505-1564}$ | NB |
| MLL4_(PHD$_5$)$_{1429-1475}$ | NB |

$K_d$ values were calculated from single measurement and errors were estimated from fitting curve by Origin software package
NB, no detectable binding
[a]ITC curves cannot be fitted reliably

**Table 3 Data collection and refinement statistics**

| | MLL3_ePHD$_6$–H4 peptide |
|---|---|
| *Data collection* | |
| Space group | *P*6 |
| Cell dimensions | |
| *a*, *b*, *c* (Å) | 85.9, 85.9, 98.7 |
| *α*, *β*, *γ* (°) | 90, 90, 120 |
| Resolution (Å) | 43.00–1.80(1.83–1.80)[a] |
| $R_{sym}$ or $R_{merge}$ | 9.8(135.7) |
| $I / \sigma I$ | 18.7(1.5) |
| Completeness (%) | 100.0(99.5) |
| Redundancy | 10.0(9.2) |
| *Refinement* | |
| Resolution (Å) | 43.00–1.80 |
| No. reflections | 36,874(1507) |
| $R_{work}/R_{free}$ | 19.4/21.6 |
| No. atoms | |
| Protein | 2766 |
| Ligand/ion | 99/14 |
| Water | 143 |
| *B*-factors | |
| Protein | 30.1 |
| Ligand/ion | 40.9/23.5 |
| Water | 33.4 |
| R.m.s. deviations | |
| Bond lengths (Å) | 0.010 |
| Bond angles (°) | 1.4 |

[a]Values in parentheses are for highest-resolution shell

important for H4-binding (Fig. 3a). The MLL3/4 complexes were purified as described previously[29]. Our western blot analysis of immunoprecipitation eluates showed that all of these purified MLL3/4 fusion proteins and their mutants equally interacted with MLL3/4's other core components, such as ASH2L, RBBP5, and WDR5, suggesting that these MLL3/4 mutations did not affect their interactions with other components (Supplementary Fig. 8).

The C terminus of MLL3 with the catalytic SET domain (MLL3_C) and its fusion with the MLL3 ePHD$_6$ domain (MLL3_ePHD$_6$-C) did not show any detectable nucleosomal methylation activity, whereas a fusion of the PHD$_{4-6}$ domain and the C terminus of MLL3 (MLL3_PHD$_{4-6}$-C) had robust methyltransferase activity (Fig. 3b), suggesting that the ePHD$_6$ domain is not sufficient for the methyltransferase activity and that other PHD domains in the PHD$_{4-6}$ cluster are also important for the enzymatic activity and functions. Single (1M: W1109A) or double (2M: W1109A/E1120A) point mutations into the PHD$_6$ domain of MLL3 reduced the enzyme activity compared to the wild-type MLL3 complex (Fig. 3b). For the MLL4 complex, the wild type MLL4 fusion protein (covering the PHD$_{4-6}$ domain and its C terminus) had robust methyltransferase activity, but a double point mutant (MLL4 fusion-2M: W1529A/E1540A) of the MLL4 fusion protein, which corresponds to the double point mutant (W1109A/E1120A) of MLL3, had weaker methyltransferase activity than did the wild-type MLL4 fusion protein (Fig. 3c). In addition, an MLL4 complex with a previously reported quadruple point mutant MLL4 (E1516A/E1517A/D1518A/E1544A) exhibited diminished activity, consistent with the previous study[29] (Fig. 3b). Of note, E1517 of MLL4 corresponds to E1097 of MLL3, which formed the salt bridge with H4R19 and thus was important for H4 binding (Fig. 2e). Taken, the ePHD$_6$-H4 interaction is essential for the H3K4 methyltransferase activity of MLL3/4.

Because our MLL3 ePHD$_6$-H4 complex structure also revealed that the H4H18-containing fragment was critical for binding to the extended PHD domain of MLL3, we generated a series of recombinant nucleosomes containing different H4 mutants to confirm the importance of these H4 residues in H3K4 methylation activity. Both MLL3 and MLL4 complexes were used in the enzymatic assays and they showed a similar trend (Fig. 3d, e). In accordance with our ITC assays and complex structure, the residues around H4H18 of histone H4 were essential for the MLL3/4's methyltransferase activity. In particular, H4H18 was critical as its single mutation (H4H18A) almost abolished the methyltransferase activity. Our pull-down assay between the GST-tagged ePHD$_6$ of MLL3/4 and wild type and mutant recombinant nucleosomes also indicated that the residues around H4H18 of histone H4 played an important role in the interaction between MLL3/4 and nucleosome, since the H18A and H4_15–19 to G mutant nucleosomes exhibited significantly reduced interaction with MLL3/4 (Supplementary Fig. 9). Interestingly, although we did not detect binding between the first 12 residues of histone H4 (H4$_{1-12}$) and the ePHD$_6$ domain of MLL3/4, our enzymatic assays showed that both MLL3 and MLL4 lost some enzymatic activity when the H4R3 residue or the first 5 residues of H4 were mutated. This scenario is in agreement with our ITC binding data and indicates that the H4R3-containing fragment may moderately interact with some negatively patched surface in ePHD$_6$. Overall, our in vitro methyltransferase assays revealed that the interaction between ePHD$_6$ of MLL3/4 and histone H4 is required for their nucleosomal methylation activity.

**MLL4-mediated NT2/D1 cell differentiation requires H4 binding**. It has been previously shown that ectopic expression of an MLL4 fusion protein containing ePHD$_6$ rescues morphological

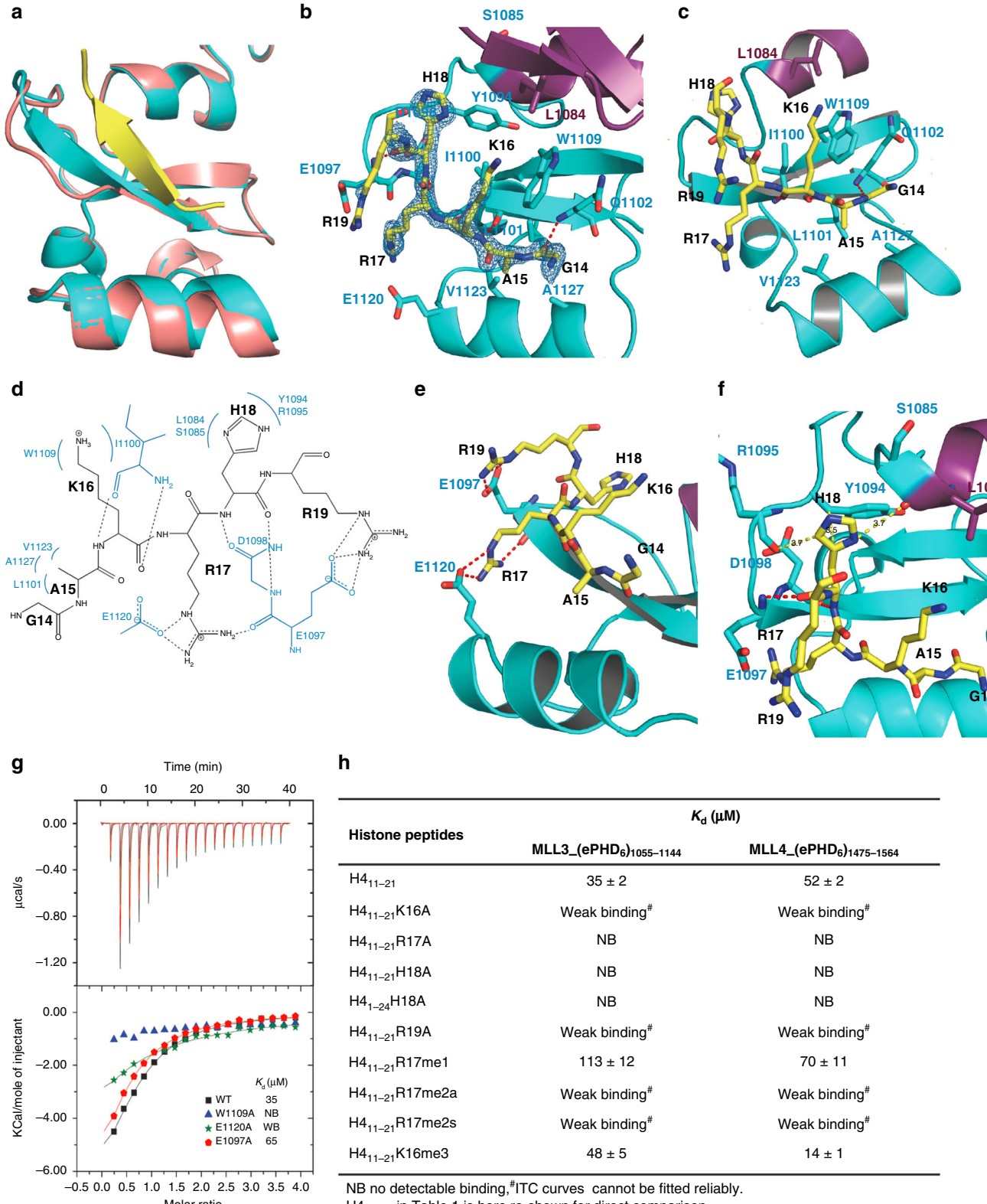

**Fig. 2** The ePHD$_6$ domain of MLL3 specifically binds to the H4H18-containing motif of histone H4. **a** Superimposition of the structures of the H4 peptide-bound and peptide-free ePHD$_6$ domain of MLL3. **b** Cartoon representation of the PHD$_6$ and preceding zinc finger colored in cyan and purple, respectively. The histone H4 peptide is shown with a Fo–Fc omit map contoured at 2.5 $\sigma$. **c**, **e**, **f** Detailed interactions between the ePHD$_6$ domain of MLL3 and residues A15/K16, R17/R19, and H18 of histone H4 peptide, respectively. The histone H4 peptide is shown in sticks and colored in yellow. Interacting residues are shown in stick mode and red/yellow dashed lines represent intra-molecular hydrogen bonds and atom distances, respectively. **d** Schematic of the detailed interactions between the ePHD$_6$ domain of MLL3 and the H4H18 histone H4 peptide. The histone H4 interacting residues of MLL3 are colored in blue. Dashed lines represent hydrogen bonds. **g** ITC curves for the titration of wild type or different mutants of ePHD$_6$ domain of MLL3 to the histone H4 peptide H4$_{11-21}$. NB no detectable binding, WB, weak binding. **h** Binding affinities of different histone peptides to ePHD$_6$ domain of MLL3 and MLL4 measured by ITC

Table h:

| Histone peptides | $K_d$ (μM) | |
| --- | --- | --- |
| | MLL3_(ePHD$_6$)$_{1055-1144}$ | MLL4_(ePHD$_6$)$_{1475-1564}$ |
| H4$_{11-21}$ | 35 ± 2 | 52 ± 2 |
| H4$_{11-21}$K16A | Weak binding[#] | Weak binding[#] |
| H4$_{11-21}$R17A | NB | NB |
| H4$_{11-21}$H18A | NB | NB |
| H4$_{1-24}$H18A | NB | NB |
| H4$_{11-21}$R19A | Weak binding[#] | Weak binding[#] |
| H4$_{11-21}$R17me1 | 113 ± 12 | 70 ± 11 |
| H4$_{11-21}$R17me2a | Weak binding[#] | Weak binding[#] |
| H4$_{11-21}$R17me2s | Weak binding[#] | Weak binding[#] |
| H4$_{11-21}$K16me3 | 48 ± 5 | 14 ± 1 |

NB no detectable binding, [#]ITC curves cannot be fitted reliably.
H4$_{11-21}$ in Table 1 is here re-shown for direct comparison.

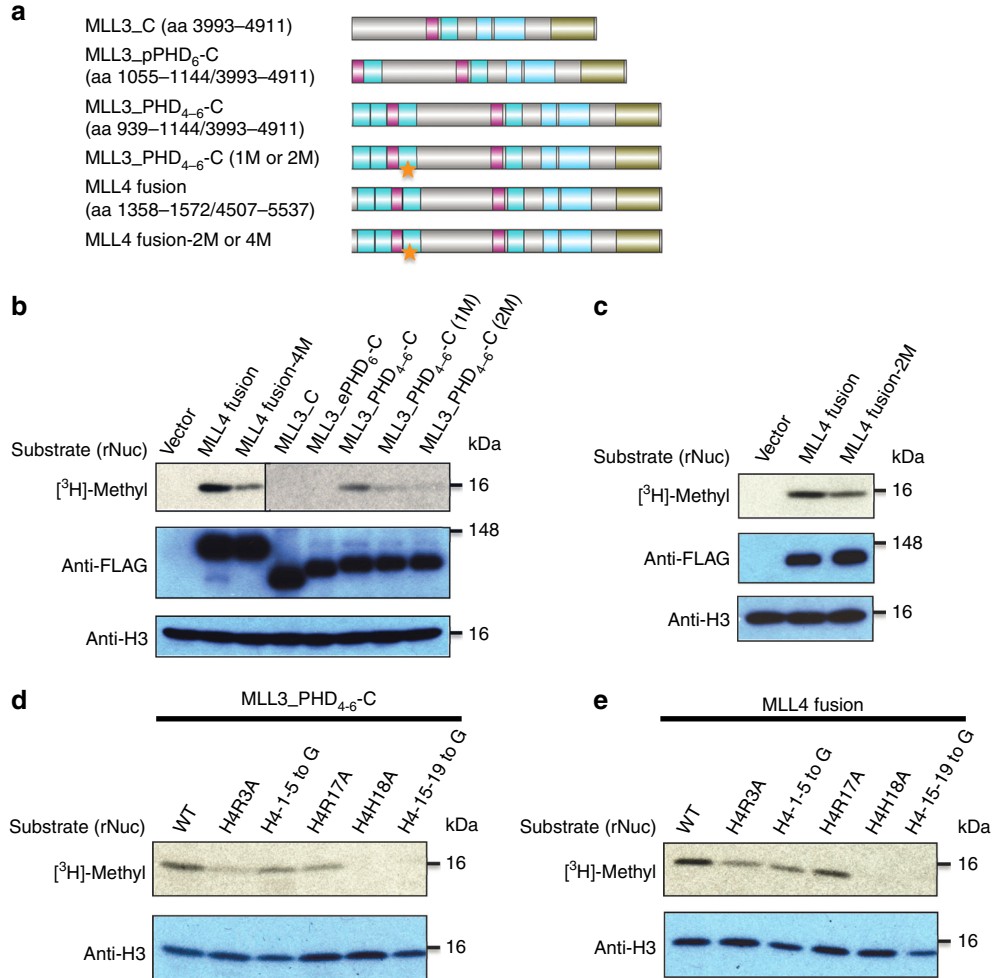

**Fig. 3** Interaction between ePHD$_6$ of MLL3/4 and histone H4 is required for their nucleosomal methylation activity. **a** Schematic representation of constructs used in the histone methyltransferase (HMT) assay. Stars indicate the mutations in PHD$_6$ of MLL3/MLL4. 1 M is W1109A single point mutation of MLL3, 2 M: W1109A/E1120A and W1529A/E1540A are double point mutations of MLL3 and corresponding residues in MLL4, respectively, MLL4 fusion-4M is a quadruple point mutant (E1516A/E1517A/D1518A/E1544A) of MLL4, and E1517 of MLL4 corresponds to E1097 of MLL3. **b, c** Comparison of HMT activities of different MLL3/MLL4 constructs. **d, e** Effects of different histone H4 mutants on HMT activities of MLL3_PHD$_{4-6}$-C and MLL4 fusion proteins

differentiation defects of MLL4-depleted NT2/D1 cells, as well as impaired expression of MLL4 target genes (e.g., the differentiation-specific genes *HOXA1–3*) in the same cells during retinoic acid (RA)-induced neuronal differentiation[29]. To determine the importance of the interaction between ePHD$_6$ and the H4H18-containing H4 region in regulating MLL4 target genes and NT2/D1 cell differentiation, we examined whether the double point mutation (2M) in the MLL4 fusion protein impeded MLL4-mediated rescue of differentiation defects of the MLL4-depleted NT2/D1 cells during RA-induced differentiation. Ectopic expression of the wild-type MLL4 fusion construct but not the MLL4 fusion-2M construct restored defective differentiation of MLL4-depleted NT2/D1 cells (Fig. 4a, b). Consistent with this, the wild-type MLL4 fusion construct but not its mutant rescued expression levels of *HOXA1–3* and the neuron-specific gene *NeuN* during RA-induced neuronal differentiation of MLL4-depleted cells (Fig. 4c). In contrast, the MLL4 fusion construct had no substantial effect on expression of the pluripotent gene *NANOG*, suggesting its specific effect on expression of differentiation-associated genes, such as *HOXA1–3* and *NeuN* (Fig. 4c). These results indicate that the interaction between the H4H18-containing H4 region and ePHD$_6$ is indispensable for MLL4-

mediated gene activation and proper morphological changes of NT2/D1 cells during RA-induced neuronal differentiation.

## Discussion

In this study, we reported a unique binding mode of the H4H18-containing histone H4 fragment by the extended PHD domain of MLL3/4. H4H18 is located in a short basic fragment of the N-terminal histone H4 tail, and this basic batch ($_{16}$KRHRK$_{20}$) has been shown to be important for recruiting and/or activating various chromatin modifying activities. For instance, this basic batch is required to target some ISWI family of chromatin remodeling complexes to their specific chromatin regions, and stimulate the ISWI ATPase activity[41,42]. The same basic patch of histone H4 could also bind to Dot1, and this interaction is essential for Dot1-mediated histone H3K79 methylation and proper telomere silencing[43,44]. Based on the crystal structure of nucleosome, the histone tails including the basic batch of histone H4 are unstructured and exposed to solvent[45], but a recent study reveals that binding of the silencing protein Sir3 (silent information regulator 3) to nucleosomes induces a conformational change in the N-terminal tail of histone H4 that promotes

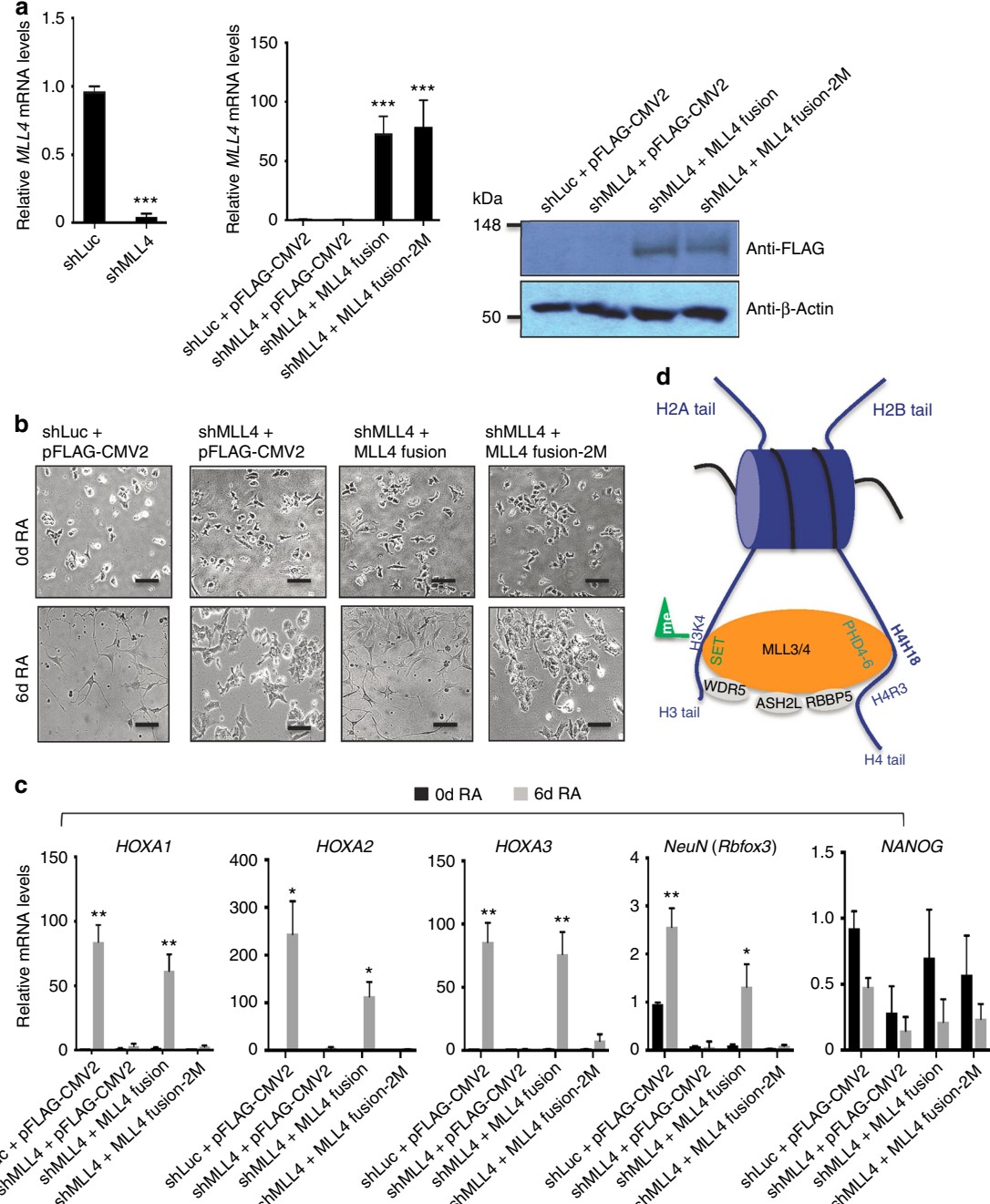

**Fig. 4** The binding activity of ePHD$_6$ is essential for MLL4-mediated gene activation and cell-morphological changes during RA-induced NT2/D1 cell differentiation. **a** The efficiency of knockdown of MLL4 by shMLL4 (left) and rescue by ectopic expression of MLL4 fusion or MLL4 fusion-2M (W1529A/E1540A) (right) were examined by quantitative RT-PCR or western blot analysis. Data are presented as the mean ± SEM (error bars) of three independent experiments. Where indicated, statistical P-values were determined using two-tailed Student's t-test. ***p < 0.001 indicate statistically significant differences. **b** The effect of ectopic expression of MLL4 fusion or MLL4 fusion-2M (W1529A/E1540A) on differentiation morphologies during RA-induced neuronal differentiation of shMLL4-treated NT2/D1 cells. Morphological changes during cellular differentiation were monitored using a microscope (10×). Shown are representative images of three independent experiments. Black scale bars, 50 μm. **c** The effect of MLL4 fusion or MLL4 fusion-2M on expression of the HOXA1–3, NeuN, and NANOG genes during RA-induced neuronal differentiation of shMLL4-treated NT2/D1 cells. Expression levels were measured by quantitative RT-PCR. Data are presented as the mean ± SEM (error bars) of three independent experiments. Where indicated, statistical P-values were determined using two-tailed Student's t-test. *p < 0.05, **p < 0.01, and ***p < 0.001 indicate statistically significant differences. **d** A possible model of the interaction between MLL3/4 and nucleosome. MLL3/4 are recruited to their targeted sites by the interaction between the PHD$_{4-6}$ (ePHD$_6$) domains and histone H4, and then the SET domains of MLL3/4 catalyze the methylation of histone H3K4 of the same or the nearby nucleosome

interactions between that basic batch of histone H4 and nucleo-somal DNA, which is critical for establishment and maintenance of silent chromatin domains at telomeres and mating type loci[46]. Therefore, the H4H18-containing basic batch of histone H4 is

involved in various chromatin modifying events, indicating its importance in chromatin biology.

The PHD domains of MLL3/4 has been reported to play an important role in recruiting the MLL3/4 complexes to their target

genes by binding to histone H4 and depositing the active H3K4me3 marks on the same genes for differentiating the NT2/D1 stem cells[29]. Our structural and binding studies revealed that an H4H18-containing fragment of histone H4 bound to the $ePHD_6$ of MLL3/4 specifically by forming a β-sheet with the two β strands of the $PHD_6$ and that other PHD domains in $PHD_{4-6}$ cluster are also important for the enzymatic activity and functions. On the other hand, our enzymatic results, displayed here and shown previously[29], revealed that the H4R3-containing fragment also contributes to the H4-binding-mediated H3K4me3 methylation, potentially by binding to some negatively charged patch of the $ePHD_6$ domain of MLL3/4 (Fig. 4d).

Both the $ePHD_6$ and $PHD_6$ domains of MLL3 behaved as a monomer in solution, but the $ePHD_6$ and $PHD_6$ domains of MLL4 behaved as a dimer in solution (Supplementary Fig. 3). Our enzymatic data showed that the $ePHD_6$ binding to histone H4 is essential but not sufficient for the MLL3/4-mediated histone H3K4 methylation activity. The $PHD_{4-6}$ domains of MLL3/4 are required for the MLL3/4-mediated histone H3K4 methylation activity. Because we are not able to obtain stable proteins containing the $PHD_{4-6}$ domain, so we are not clear about the polymerization states of the $PHD_{4-6}$ domains or even the full-length MLL3/4 proteins. It needs to further characterize in the future if and how the polymerization states of the $PHD_{4-6}$ domains or even the full-length MLL3/4 proteins affect their enzymatic and biological functions.

The histone H4 N-terminal tail is subject to various post-translation modifications, such as H4R3 methylation and H4K5/8/12 acetylation. Several post-translational modifications have been observed in the H4H18-containing fragment, such as the well-studied histone H4K16 acetylation, H4K20 methylation, and the less-studied H4K16 methylation[34], H4R17 and H4R19 methylation[34]. It has been shown that PRMT7-regulated H4R3 symmetric dimethylation would hamper its binding to the PHD domains of MLL4, and repress MLL4 target genes[29]. Interestingly, PRMT7 could also methylate histone H4R17[34,35]. Histone H3K4 methylation by another SET1/MLL family of methyltransferase MLL1 was reported to be trans-histone regulated, i.e., MLL1-mediated histone H3K4 methylation acts synergistically with histone H4K16 acetylation by the histone acetyltransferase MOF for optimal transcription activation in their target genes[47]. Our data showed that methylation of H4R17 diminished its binding to both MLL3 and MLL4, supporting the notion that MLL3/4 and PRMT7 exhibits opposing effects on cellular differentiation[29] and suggesting the molecular basis of another trans-tail regulation mechanism for MLL3/4-mediated H3K4 methylation by H4R17 methylation. Furthermore, our data also showed that methylation of H4K16 modestly increased its binding to MLL4, but not to MLL3, which may further introduce elaborate regulatory mechanism between MLL3 and MLL4 functions. However, this mechanism needs further investigation in the future.

## Methods

**Plasmids, cell lines, and antibodies.** The *E coli* expression vector pET28GST-LIC was constructed by ourselves (GenBank accession EF456739, https://www.ncbi.nlm.nih.gov/nuccore/134105587/) and the mammalian expression vector pFLAG-CMV2 (Sigma-Aldrich, E7033) was purchased from Sigma-Aldrich Company. The small hairpin (sh) MLL4 (Sigma-Aldrich, SHCLND-NM_003482) and shLuc (shLuciferase, Sigma-Aldrich, SHC007) plasmids were purchased from Sigma-Aldrich company. The lentiviral packaging plasmids such as pCMV-deltaR8.2 (A14C, E45C) (Addgene, 79047) and pCMV-VSV-G (Addgene, 8454) were purchased from Addgene.

The NT2/D1 (ATCC® CRL-1973) embryonic carcinoma cell line and HEK 293T (ATCC® ACS-4500™) embryonic renal cell line were purchased from ATCC and maintained in Dulbecco's modified Eagle's medium (DMEM) supplemented with 10% fetal bovine serum. These cell lines were authenticated by short tandem repeat (STR) profile method and tested negative for mycoplasma contamination by PCR.

Antibodies used in this research are as follows: mouse monoclonal anti-FLAG M2 antibody (Sigma-Aldrich, F3165, 1:1000), mouse monoclonal anti-β-actin antibody (Sigma-Aldrich, A5441, 1:1000), rabbit polyclonal anti-H3 antibody (Abcam, ab1791, 1:1000), mouse monoclonal anti-H4 antibody (Abcam, ab174628, 1:1000), rabbit polyclonal anti-ASH2L antibody (Bethyl Laboratories, A300-107A, 1:2000), rabbit polyclonal anti-RBBP5 antibody (Bethyl Laboratories, A300-109A, 1:2000), and rabbit polyclonal anti-WDR5 antibody (Millipore Corporation, 07-706, 1:4000).

**Protein expression and purification.** The DNA fragments of MLL3 and MLL4 PHD domains (MLL3_$ePHD_6$, residues 1055–1144 and residues 1075–1144; MLL3_$PHD_6$, residues 1085–1144; MLL3_$PHD_5$, residues 1009–1055; MLL3_$PHD_{5-6}$, residues 1008–1144; MLL3_$ePHD_3$, residues 436–525; MLL4_$ePHD_6$, residues 1475–1564, and residues 1495–1564; MLL4_$PHD_6$ residues 1505–1564 and MLL4_$PHD_5$ residues 1429–1475) were subcloned into a modified pET28GST-LIC vector by T4 ligase-independent method (Clontech, 638920) to generate N-terminal GST, His-tagged fusion protein. The recombinant protein was over-expressed in *E coli* BL21 (DE3) Codon plus RIL (Stratagene) at 15 °C by induction with 0.25 mM IPTG at an $OD_{600}$ of 0.8 and purified by affinity chromatography on Ni-nitrilotriacetate resin (Qiagen, 30250) followed by thrombin protease treatment to remove the tag. The protein was further purified by HiTrap Q HP column (GE Healthcare, 17115401) by using buffers with 20 mM Tris, pH 8.0, 1 mM DTT, 50 mM NaCl (low salt buffer) or 1 M NaCl (high salt buffer), and Superdex75 gel-filtration column (GE Healthcare, 28989333) by using a buffer containing 20 mM Tris, pH 7.5, 150 mM NaCl, 50 μM $ZnCl_2$, and 1 mM DTT. For crystallization experiments, purified protein was concentrated to 5 mg mL$^{-1}$ in the same buffer as gel-filtration by using Amicon Ultra-15 Centrifugal Filter Units (Millipore Corporation, UFC901024).

Mammalian expression plasmids encoding different fusions of MLL3 PHD and SET domains were subcloned into the pFLAG-CMV2 vector (Sigma-Aldrich, E7033) and were transiently expressed in HEK 293T cells (ATCC® ACS-4500™) by using Lipofectamine 2000 (ThermoFisher Scientific, 11668027) according to manufacturer's protocol. Two days after transfection, cells were harvested and then lysed by mammalian lysis buffer (20 mM Tris-HCl, 137 mM NaCl, 1.5 mM $MgCl_2$, 1 mM EDTA, 10% glycerol, 1% Triton X-100, 0.2 mM PMSF, 1 μg mL$^{-1}$ aprotinin, 2.5 μg mL$^{-1}$ leupeptin, and 1 μg mL$^{-1}$ pepstatin at pH 8.0). FLAG immunoprecipitation (FLAG IP) was performed in a similar way as described previously[29]. In brief, total cell lysates were incubated with anti-FLAG M2 Affinity Gel (Sigma-Aldrich, A2220) in 4 °C for 5 h and were extensively washed with BC500 (20 mM Tris-HCl, 500 mM KCl, 1.5 mM $MgCl_2$, 0.2 mM EDTA, 10% glycerol, 0.2 mM PMSF at pH 8.0). The FLAG-tagged proteins were eluted by using 0.4 μg mL$^{-1}$ of FLAG peptides in HMT buffer (50 mM Tris-HCl, 100 mM KCl, 5 mM $MgCl_2$, 4 mM DTT and 10% glycerol at pH 8.5) and were used for further analysis. MLL4 fusion and MLL4 fusion-4M (previously described as MLL4$_{fusion}$ and mMLL4$_{fusion}$, respectively) have been reported[29]. MLL4 fusion-2M was generated by mutating MLL4 fusion[29] using QuickChange Site-Directed Mutagenesis Kit (Stratagene, 200518) according to manufacturer's instruction. All the FLAG-tagged MLL4 proteins were purified as the MLL3 constructs above.

Recombinant nucleosome was reconstituted as described previously with minor modifications[48,49]. Briefly, the octamer were overexpressed and induced with 0.25 mM IPTG at 15 °C and purified by affinity chromatography on Ni-nitrilotriacetate resin (Qiagen, 30250). Recombinant nucleosome was reconstituted by mixing purified octamer with 147-bp double strand DNA at a molar ratio 1.2:1 in a buffer containing 20 mM Tris, pH 7.5, 1 mM EDTA, 1 mM DTT, and 2 M KCl and dialyzing gradually to remove the salt to a final concentration of 0.01 M KCl. All the mutations were introduced with the QuikChange II XL site-directed mutagenesis kit (Stratagene, 200522) and confirmed by DNA sequencing. Mutants were overexpressed and purified as the wild-type constructs above. All the primers used in this research were shown in Supplementary Table 1.

**Isothermal titration calorimetry.** For the ITC measurement, the concentrated proteins were diluted into 20 mM Tris, pH 7.5, 150 mM NaCl; the lyophilized peptides (Peptide 2.0 Inc.) were dissolved in the same buffer, and the pH value was adjusted by adding NaOH. Peptides concentrations were estimated from the mass. All the measurements were performed in duplicate at 25 °C, using a VP-ITC microcalorimeter or iTC-200 microcalorimeter (MicroCal, Inc.). The protein with a concentration of 50–100 μM was placed in the cell chamber, and the peptides with a concentration of 1–2 mM in syringe was injected in 25 (19 for iTC-200) successive injections with a spacing of 180 s (150 s for iTC-200) and a reference power of 13 μcal s$^{-1}$ (6 μcal s$^{-1}$ for iTC-200). Control experiments were performed under identical conditions to determine the heat signals that arise from injection of the peptides into the buffer. Data were fitted using the single-site binding model within the Origin software package (MicroCal, Inc.). iTC-200 data should be consistent with those from VP-ITC instrument, based on ITC results of same PHD domain using the two instruments.

**Protein crystallization.** For cocrystallization, purified proteins were mixed with different length histone H4 peptides at a molar ratio 1:3 and crystallized using the sitting drop vapor diffusion method at 18 °C by mixing 0.5 μL of the protein with 0.5 μL of the reservoir solution. The complex of $ePHD_6$ of MLL3 and histone

H4 peptide (residues 1–20) crystallized in a buffer containing 2 M sodium formate, 0.1 M Tris, pH 8.5. Before flash-freezing crystals in liquid nitrogen, crystals were soaked in a cryoprotectant consisting of 85% reservoir solution and 15% glycerol.

**Data collection and structure determination**. The diffraction data of the complex crystal of ePHD$_6$ of MLL3 and histone H4 peptide (residues 1–20) were collected at beamline CMCF 08ID-1 of Canadian Light Source (CLS) at 100 K and wavelength of 0.97949 Å. The data set was processed using the HKL-3000 suite[50]. The structure was solved by molecular replacement using MOLREP[51] using another low resolution MLL3 structure as a search template, which was solved by single-wavelength anomalous dispersion phasing method by taking advantage of the Zn ions in the ePHD$_6$ domain of MLL3. REFMAC was used for structure refinement[52]. Graphics program COOT was used for model building and visualization[53]. MOLPROBITY was used for structure validation and Ramachandran statistics calculation[54]. The Ramachandran statistics shows that 97.0% of all the residues are in the favored region and all the other residues are in the allowed region. Crystal diffraction data and refinement statistics for the structure were displayed in Table 3.

**In vitro histone methyltransferase (HMT) assay**. HMT assay was performed according to a previously described method with minor modifications[29]. Briefly, FLAG IP eluates of FLAG-tagged MLL3/4 PHD and SET fusion proteins were mixed with 0.5 μg of substrates (wild type or mutated recombinant nucleosomes) and 2.5 μCi of [$^3$H]-labeled $S$-adenosyl-ʟ-methionine ([$^3$H]-SAM, PerkinElmer, NET155V001MC). All reactions were performed in HMT buffer with a final volume of 20 μL. After incubation at 30 °C for 16 h, the reactions were terminated by adding 2xSDS sample buffer and were subjected to SDS-PAGE. The signals for HMT activity were detected by autoradiography. The input of the MLL3/4 proteins and nucleosomes were detected by mouse monoclonal anti-FLAG M2 (Sigma-Aldrich, F3165, 1:1000) and rabbit polyclonal anti-H3 antibody (Abcam, ab1791, 1:1000), respectively. All the uncropped scans of these western blot were shown in Supplementary Figs. 10–12.

**Pull-down assay**. The purified GST-tagged fusion ePHD$_6$ of MLL3 and MLL4 (10 μg) were bound to Glutathione Sepharose 4B (GE Healthcare, 28952360) for 1 h at 4 °C. After washing three times with buffer containing 20 mM Tris, pH 7.5, 150 mM NaCl, and 0.1% Triton X-100, the bound GST-tagged fusion proteins were incubated with purified recombinant wild type and mutant nucleosomes (30 μg) for another 1 h at 4 °C. After washing three times with the same buffer, the pull-down samples were eluted by adding 1xSDS sample buffer. Then the pull-down samples were detected by weston blot analysis.

**Western blot analysis**. Input and pull-down protein samples were run on NuPAGE 4–12% Bis-Tris protein gel in MOPS buffer. Proteins were transferred onto PVDF membrane and blocked overnight in 3% BSA in PBST. Membranes were incubated in mouse anti-H4 (Abcam, ab174628, 1:1000), which is generated by using the C terminus of histone H4 as antigen, for 1 h followed by three washes of 10 min in PBST. This was repeated with secondary antibody, IRDye® 680RD goat anti-mouse IgG (LI-COR, 926-68070, 1:5000). Membrane was visualised on an Odyssey® CLx Imaging System (LI-COR). The uncropped scans of this assay was shown in Supplementary Fig. 13.

**NT2/D1 cells differentiation and rescue experiments**. For the MLL4 knock-down, NT2/D1 cells were infected with lentiviruses containing shMLL4 as described previously[29]. Briefly, shMLL4 (Sigma-Aldrich, SHCLND-NM_003482) or control shLuc (shLuciferase, Sigma-Aldrich, SHC007) was cotransfected along with pCMV-deltaR8.2 (packing plasmid, Addgene, 79047) and pCMV-VSV-G (envelope plasmid, Addgene, 8454) plasmids into HEK 293T cells by using a calcium phosphate method to produce the lentivirus. Virus particles were harvested 2 days later and used to infect NT2/D1 cells for 2 days under the selection by 2.5 mg mL$^{-1}$ puromycin (the resistant marker of the shRNA plasmids). The knockdown efficiency of MLL4 was examined by quantitative RT-PCR. For the rescue experiments, the MLL4 knockdown NT2/D1 cells ($1–2 \times 10^3$) were seeded in 6-well plates, incubated for 24 h, and transfected with 5 μg of pFLAG-CMV2 expression plasmids encoding the MLL4 fusion or MLL4 fusion-2M (W1529A/ E1540A) using Fugene 6 (Roche, 11815091001). As a control, shLuc-treated cells were transfected with the empty vector pFLAG-CMV2. After 72 h incubation, the cells were treated with 10 μM RA for 6 days and harvested for further analysis. Morphological changes during cellular differentiation were monitored using a microscope (10×). Total RNAs were isolated and cDNA was synthesized. The mRNA expression levels were quantified using CFX Manager software and were normalized to 18S RNA. The relative mRNA level represents the fold change over the control. RT-PCR primer sequences for HOXA1–A3 and NANOG are the same as described earlier[55] and NeuN (Rbfox3) primers for quantitative RT-PCR were shown in the Supplementary Table 1. Data are presented as the mean ± SEM (error bars). Where indicated, statistical $P$-values were determined using two-tailed Student's $t$-test. *$p < 0.05$, **$p < 0.01$, and ***$p < 0.001$ indicate statistically significant differences.

**Data availability**

Coordinates and structure factors are deposited in the Protein Data Bank (PDB) with accession code 6MLC. All other relevant data supporting the key findings of this study are available within the Article and its Supplementary Information files or from the corresponding authors upon reasonable request. A Reporting Summary for this Article is available as a Supplementary Information file.

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

## Acknowledgements

We would like to thank Dr. Wolfram Tempel for data collection and Mengqi Zhou for the pull-down experiments. We acknowledge the use of beamline CMCF 08ID-1 of Canadian Light Source (CLS). The SGC is a registered charity (number 1097737) that receives funds from AbbVie, Bayer Pharma AG, Boehringer Ingelheim, Canada Foundation for Innovation, Eshelman Institute for Innovation, Genome Canada through Ontario Genomics Institute [OGI-055], Innovative Medicines Initiative (EU/EFPIA) [ULTRA-DD grant number 115766], Janssen, Merck KGaA, Darmstadt, Germany, MSD, Novartis Pharma AG, Ontario Ministry of Research, Innovation and Science (MRIS), Pfizer, São Paulo Research Foundation-FAPESP, Takeda, and Wellcome (J.M). This study was also supported by grants to Y.L. (31500613) and S.Q. (31500615) from the National Natural Science Foundation of China and by grants to M.G.L. from the NIH (R01CA207098 and R01CA207109) and the Center for Cancer Epigenetics at MD Anderson.

## Author contributions

Y.L. purified and crystallized the protein; Y.L. and S.Q. conducted the ITC assays; T.-Y.C. conducted the HMT assays; M.L. reconstituted the recombinant nucleosome, and conducted some ITC binding and nucleosome pull-down assays; S.S.D. performed NT2/D1 cell experiments; J.C.H. performed western blot experiments; A.D. determined the crystal structure; P.L. and Y.L. cloned the constructs; J.M. conceived and designed the study; and M.G.L. and J.M. supervised experiments. J.M. wrote the paper with substantial contributions from all the other authors.

## Additional information

**Competing interests:** The authors declare no competing interests.

