## [Peer Review File · Nature Communications]

Reviewer #1 (Remarks to the Author):

In this manuscript, Liu et al carried out crystallographic and biochemical studies of the ePHD6 of MLL3 and MLL4 and their interactions with the histone peptides of H4. They solved the crystal structure of the complex of MLL3 ePHD6 and its binding peptide, and identified that the H4H18-containing region of H4 is the major binding site of MLL3/4 ePHD6. They further made a series of mutations to verify the contribution of related residues on both MLL3 ePHD6 and H4, and identified key residues important for their interaction. PHD fingers have been extensively studied as the chromatin-associated domains since they were identified to bind histones with or without modifications, however, that the H4(16-20) region as the recognition site to recruit PHD-containing proteins was first described in this manuscript, which may contribute to an important function of a subsets of the PHD-finger containing proteins, especially, the MLL3/4. To identify the function of this H4H18-specific recruitment by MLL3/4 ePHD6, they executed histone methyltransferase assays and found that the interaction to H4 by ePHD6 is important for the histone methyltransferase activity of MLL3/4 on the nucleosome substrate. Taken together, they identify a new recruitment mechanism of MLL3/4 through ePHD6, which could be an important part for the full function of MLL3/4. The findings identified in this manuscript are completely new, and may have important implications for understanding the enhancer-specific methylation or recruitment by MLL3/4.

However, several major and minor issues need to be clarified to further improve the manuscript.

The first major issue, the authors show that ePHD6 binds the H4H18-containing region, then could other parts of the histones be recognized by ePHD6 of MLL3/4? In this story, the ePHD6/H4 binding was shown to be related to MLL3/4 methyltransferase activity, how these two events correlated? The authors seem to implicate that the recruitment of MLL3/4 to the nucleosome substrate by the ePHD6 will be needed for the methylation. To prove that, the authors need to show that ePHD6 binds the wildtype but not the H4H18A-mutated nucleosome by pulldown assay or other methods. If true, this may also help to show that ePHD6 binds only to the H4H18-containing region, but not other parts of the nucleosome.

Second, the oligomeric state is important for the function of an enzyme. The swapped zinc finger seems to be introduced by crystal packing, as in gel filtration, MLL3 ePHD6 turns to be a monomer. Is the same region a monomer in MLL4, could this region be crystalized?

Several minor issues are listed below.

First, based on your study, could you predict whether there are other ePHD fingers that could bind the H4H18-containing region? Is this H4-binding unique to MLL3 and MLL4 ePHD6, or is this a more general mechanism for a subset of PHD fingers?

Second, could you show the electron density of the H4 peptide in your complex structure?

Third, in the crystallographic table, why Rwork is higher than Rfree?

Fourth, why Rmerge at the highest resolution shell is smaller than the overall Rmerge?

Fifth, what is the $I/\delta I$ at the highest resolution shell?

Reviewer #2 (Remarks to the Author):

In this manuscript, Liu and coworkers clarify the functional interaction of MLL3 and MLL4 methyltransferases with histone H4. They demonstrate that the sixth PHD finger of MLL3 and MLL4 extended at the N-terminus by a zinc finger motif (ePHD6) specifically recognizes histone H4. The X-ray structure of MLL3 ePHD6 in complex with an H4 peptide reveals 6 well-defined H4 residues (G14-R19) in contact with the PHD6 region of ePHD6. The structure was validated by mutagenesis and histone H4 peptide binding affinity measurements using ITC. It is notably shown that H4 H18 is critical for the ePHD6/H4 interaction. The authors also investigate the functional importance of this interaction in the context of reconstituted MLL3 and MLL4 complexes. They show that abolishing the ePHD6/H4 interaction by introducing mutations in ePHD6 or H4 diminishes the H3K4 nucleosomal methyltransferase activities of MLL3 and MLL4 in vitro.

The current work is of excellent quality. However, this referee feels that additional functional assays with MLL3 or MLL4 in a cellular context would strengthen the manuscript for publication in Nature Communications.

Reviewer #1 (Remarks to the Author):

In this manuscript, Liu et al carried out crystallographic and biochemical studies of the ePHD6 of MLL3 and MLL4 and their interactions with the histone peptides of H4. They solved the crystal structure of the complex of MLL3 ePHD6 and its binding peptide, and identified that the H4H18-containing region of H4 is the major binding site of MLL3/4 ePHD6. They further made a series of mutations to verify the contribution of related residues on both MLL3 ePHD6 and H4, and identified key residues important for their interaction. PHD fingers have been extensively studied as the chromatin-associated domains since they were identified to bind histones with or without modifications, however, that the H4(16-20) region as the recognition site to recruit PHD-containing proteins was first described in this manuscript, which may contribute to an important function of a subsets of the PHD-finger containing proteins, especially, the MLL3/4. To identify the function of this H4H18-specific recruitment by MLL3/4 ePHD6, they executed histone methyltransferase assays and found that the interaction to H4 by ePHD6 is important for the histone methyltransferase activity of MLL3/4 on the nucleosome substrate. Taken together, they identify a new recruitment mechanism of MLL3/4 through ePHD6, which could be an important part for the full function of MLL3/4. The findings identified in this manuscript are completely new, and may have important implications for understanding the enhancer-specific methylation or recruitment by MLL3/4. However, several major and minor issues need to be clarified to further improve the manuscript.

The first major issue, the authors show that ePHD6 binds the H4H18-containing region, then could other parts of the histones be recognized by ePHD6 of MLL3/4? In this story, the ePHD6/H4 binding was shown to be related to MLL3/4 methyltransferase activity, how these two events correlated? The authors seem to implicate that the recruitment of MLL3/4 to the nucleosome substrate by the ePHD6 will be needed for the methylation. To prove that, the authors need to show that ePHD6 binds the wildtype but not the H4H18A-mutated nucleosome by pulldown assay or other methods. If true, this may also help to show that ePHD6 binds only to the H4H18-containing region, but not other parts of the nucleosome.

Response: Thanks for the reviewer's suggestion. In this revision, we added the nucleosome pulldown data (Fig. S9 and also attached below). Both our ITC binding and enzyme activity assays indicated the H4H18-containing fragment of histone H4 is critical for the interaction between MLL3/4 and nucleosome, and MLL3/4's methyltransferase activity. Our pull-down assay between the GST-tagged fusion ePHD₆ of MLL3/4 and wild type and mutant recombinant nucleosomes also indicated that the residues around H4H18 of histone H4 played an important role in the interaction between MLL3/4 and nucleosome, since the H18A and H4_15-19 to G

mutants have significantly weakened interaction with MLL3/4 (Supplementary Fig. 9). On the other hand, the ePHD₆ domains of MLL3 and MLL4 also exhibited slightly higher binding affinity to the H4₁₋₂₄ peptide compared to the H4₁₁₋₂₁ peptide, and the R3A mutant of the histone H4₁₋₂₄ peptide displayed reduced binding affinity to ePHD₆ of MLL3 and MLL4 by 2 or 3 folds, respectively, indicating that the H4R3-containing fragment somewhat may contribute to the MLL3/4 binding, probably through non-specific electrostatic interactions. We added this into our revision as well.

Supplementary Figure 9 Pull-down assay between GST-tagged fusion ePHD₆ of MLL3/4 and wild type and mutant recombinant nucleosomes detected by Western blot analysis by using anti-H4 specific antibody (Abcam, ab174628), which is generated by using the C-terminus of histone H4 as antigen.

Second, the oligomeric state is important for the function of an enzyme. The swapped zinc finger seems to be introduced by crystal packing, as in gel filtration, MLL3 ePHD₆ turns to be a monomer. Is the same region a monomer in MLL4, could this region be crystalized?

Response: Both the ePHD₆ and PHD₆ domains of MLL3 behaved as a monomer in solution, but the ePHD₆ and PHD₆ domains of MLL4 behaved as a dimer in solution (Supplementary Fig. 3). Our enzymatic data showed that the ePHD₆ binding to histone H4 is essential but not sufficient for the MLL3/4-mediated histone H3K4 methylation activity. Instead, the PHD₄₋₆ domains of MLL3/4 are required for the MLL3/4-mediated histone H3K4 methylation activity. Because we are not able to obtain stable proteins containing the PHD₄₋₆ domain, so we are not clear about the polymerization states of the PHD₄₋₆ domains or even the full-length MLL3/4 proteins. It needs further characterization in the future on if and how the polymerization states of the PHD₄₋₆ domains or even the full-length MLL3/4 proteins affect their enzymatic and biological functions. We showed the gel filtration data as a supplemental figure and added an explanation in the Discussion Section.

Supplementary Figure 3 Gel filtration chromatography of MLL3 ePHD₆ (red), PHD₆ (pink), MLL4 ePHD₆ (brown), PHD₆ (green) and protein molecular weight standards (blue, Bio-Rad). Samples were loaded to Superdex200 10/300 GL (GE Healthcare) in the buffer containing 20 mM Tris, pH 7.5, 150 mM NaCl, 50 μM ZnCl₂, and 1 mM DTT.

We tried to crystallize the ePHD₆ of MLL4 protein, however we failed to obtain crystals.

Several minor issues are listed below.

First, based on your study, could you predict whether there are other ePHD fingers that could bind the H4H18-containing region? Is this H4-binding unique to MLL3 and MLL4 ePHD₆, or is this a more general mechanism for a subset of PHD fingers?

Response: Our sequence blast just revealed 3 similar ePHD domains: 2 (ePHD₃ and ePHD₆) from MLL3 and one (ePHD₆) from MLL4. Our ITC binding data indicate that only the ePHD₆ of MLL3/4 could bind to H4H18-containing peptides. The ePHD₃ of MLL3, which share higher similarity to ePHD₆ of MLL3 (82%, Fig. 1B), could not bind to histone H4₁₁₋₂₁ peptide. Hence, this interaction may be not as general as the H3K4 binding PHD domains. So far, most solved PHD fingers utilize an enclosed

binding pocket to recognize the free N-terminal amine group of residue A1 of histone H3, whereas the MLL3 ePHD₆ domain has an open binding groove, which explains why the MLL3 ePHD₆ domain is able to bind to a sequence motif in the middle of the histone H4 N-terminal tail (Supplementary Fig. 5 and 6).

Second, could you show the electron density of the H4 peptide in your complex structure?

Response: We had added the electron density of the H4 peptide in revised Fig. 2B.

Third, in the crystallographic table, why Rwork is higher than Rfree?

Response: We fixed the mistake.

Fourth, why Rmerge at the highest resolution shell is smaller than the overall Rmerge?

Response: We fixed the mistake.

Fifth, what is the I/σI at the highest resolution shell?

Response: 1.5

Reviewer #2 (Remarks to the Author):

In this manuscript, Liu and coworkers clarify the functional interaction of MLL3 and MLL4 methyltransferases with histone H4. They demonstrate that the sixth PHD finger of MLL3 and MLL4 extended at the N-terminus by a zinc finger motif (ePHD6) specifically recognizes histone H4. The X-ray structure of MLL3 ePHD6 in complex with an H4 peptide reveals 6 well-defined H4 residues (G14-R19) in contact with the PHD6 region of ePHD6. The structure was validated by mutagenesis and histone H4 peptide binding affinity measurements using ITC. It is notably shown that H4 H18 is critical for the ePHD6/H4 interaction. The authors also investigate the functional importance of this interaction in the context of reconstituted MLL3 and MLL4 complexes. They show that abolishing the ePHD6/H4 interaction by introducing mutations in ePHD6 or H4 diminishes the H3K4 nucleosomal methyltransferase activities of MLL3 and MLL4 in vitro.

The current work is of excellent quality. However, this referee feels that additional functional assays with MLL3 or MLL4 in a cellular context would strengthen the manuscript for publication in Nature Communications.

Response: Thanks for the reviewer's suggestion. In this revision, we have added the

cellular experiments and our data indicated that the binding activity of ePHD₆ is required for MLL4-mediated neuronal differentiation of NT2/D1 cells. Please check our Figure 4 and corresponding section for details.

Figure 4. The binding activity of ePHD₆ is essential for MLL4-mediated gene activation and cell-morphological changes during RA-induced NT2/D1 cell differentiation. (A and B) The effect of ectopic expression of MLL4 fusion or MLL4 fusion-2M (W1529A/E1540A) on differentiation morphologies during RA-induced neuronal differentiation of shMLL4 #5 (shMLL4)-treated NT2/D1 cells. MLL4 fusion or MLL4 fusion-2M (W1529A/E1540A) was transfected using Fugene 6-mediated transfection, and their expression levels were assessed by quantitative RT-PCR and

Western blot analysis (A). Three days later, cells ($1-2 \times 10^3$) were seeded onto 6-well plates and treated with 10 μ M RA for 6 days. Morphological changes during cellular differentiation were monitored using a microscope (10x); Shown are representative images of three independent experiments (B). shLuc (shLuciferase)-treated cells transfected with the empty vector pFLAG-CMV2 (shLuc + pFLAG-CMV2) was used as a control. Black scale bars, 50 μ m. (C) The effect of MLL4 fusion or MLL4 fusion-2M on expression of the *HOXA1-3*, *NeuN*, and *NANOG* genes during RA-induced neuronal differentiation of shMLL4 #5 (shMLL4)-treated NT2/D1 cells. Expression levels were measured by quantitative RT-PCR. Data are presented as the mean \pm SEM (error bars) of three independent experiments. (D) A possible model of the interaction between MLL3/4 and nucleosome. MLL3/4 are recruited to their targeted sites by the interaction between the PHD₄₋₆ (ePHD₆) domains and histone H4, then the SET domains of MLL3/4 catalyze the methylation of histone H3K4 of the same or the nearby nucleosome.

Reviewer #1 (Remarks to the Author):

The authors have performed additional experiments that addressed all my concerns. I think this is a very nice structural and biochemical study on an important new finding. I support its publication.

Reviewer #2 (Remarks to the Author):

The new data presented in Figure 4 (and pages 14 and 15) nicely show that the binding activity of ePHD6 is needed for MLL4-mediated differentiation. This new finding strengthens the manuscript. The manuscript is ready for publication.